# Characterization and Genome Study of a Newly Isolated Temperate Phage Belonging to a New Genus Targeting *Alicyclobacillus acidoterrestris*

**DOI:** 10.3390/genes14061303

**Published:** 2023-06-20

**Authors:** Dziyana Shymialevich, Michał Wójcicki, Olga Świder, Paulina Średnicka, Barbara Sokołowska

**Affiliations:** 1Culture Collection of Industrial Microorganisms—Microbiological Resources Center, Department of Microbiology, Prof. Wacław Dąbrowski Institute of Agricultural and Food Biotechnology—State Research Institute, Rakowiecka 36 Str., 02-532 Warsaw, Poland; diana.szymielewicz@ibprs.pl; 2Laboratory of Biotechnology and Molecular Engineering, Department of Microbiology, Prof. Wacław Dąbrowski Institute of Agricultural and Food Biotechnology—State Research Institute, Rakowiecka 36 Str., 02-532 Warsaw, Poland; michal.wojcicki@ibprs.pl (M.W.); paulina.srednicka@ibprs.pl (P.Ś.); 3Department of Food Safety and Chemical Analysis, Prof. Wacław Dąbrowski Institute of Agricultural and Food Biotechnology—State Research Institute, Rakowiecka 36 Str., 02-532 Warsaw, Poland; olga.swider@ibprs.pl; 4Department of Microbiology, Prof. Wacław Dąbrowski Institute of Agricultural and Food Biotechnology—State Research Institute, Rakowiecka 36 Str., 02-532 Warsaw, Poland

**Keywords:** bacteriophages, *Alicyclobacillus acidoterrestris*, whole-genome sequencing, genomic analysis, functional annotation, transmission-electron microscopy, food biocontrol

## Abstract

The spoilage of juices by *Alicyclobacillus* spp. remains a serious problem in industry and leads to economic losses. Compounds such as guaiacol and halophenols, which are produced by *Alicyclobacillus*, create undesirable flavors and odors and, thus, decrease the quality of juices. The inactivation of *Alicyclobacillus* spp. constitutes a challenge because it is resistant to environmental factors, such as high temperatures, and active acidity. However, the use of bacteriophages seems to be a promising approach. In this study, we aimed to isolate and comprehensively characterize a novel bacteriophage targeting *Alicyclobacillus* spp. The *Alicyclobacillus* phage strain KKP 3916 was isolated from orchard soil against the *Alicyclobacillus acidoterrestris* strain KKP 3133. The bacterial host’s range and the effect of phage addition at different rates of multiplicity of infections (MOIs) on the host’s growth kinetics were determined using a Bioscreen C Pro growth analyzer. The *Alicyclobacillus* phage strain KKP 3916, retained its activity in a wide range of temperatures (from 4 °C to 30 °C) and active acidity values (pH from 3 to 11). At 70 °C, the activity of the phage decreased by 99.9%. In turn, at 80 °C, no activity against the bacterial host was observed. Thirty minutes of exposure to UV reduced the activity of the phages by almost 99.99%. Based on transmission-electron microscopy (TEM) and whole-genome sequencing (WGS) analyses, the *Alicyclobacillus* phage strain KKP 3916 was classified as a tailed bacteriophage. The genomic sequencing revealed that the newly isolated phage had linear double-stranded DNA (dsDNA) with sizes of 120 bp and 131 bp and 40.3% G+C content. Of the 204 predicted proteins, 134 were of unknown function, while the remainder were annotated as structural, replication, and lysis proteins. No genes associated with antibiotic resistance were found in the genome of the newly isolated phage. However, several regions, including four associated with integration into the bacterial host genome and excisionase, were identified, which indicates the temperate (lysogenic) life cycle of the bacteriophage. Due to the risk of its potential involvement in horizontal gene transfer, this phage is not an appropriate candidate for further research on its use in food biocontrol. To the best of our knowledge, this is the first article on the isolation and whole-genome analysis of the *Alicyclobacillus*-specific phage.

## 1. Introduction

Fruit juices are among the most popular beverages and play an important role in human health due to their nutrient content and bioactive compounds [1,2]. They are low-calorie drinks rich in nutrients and bioactive compounds, such as vitamins, proteins, carbohydrates, polyphenols, minerals, enzymes, fiber, and antioxidants [3,4,5]. According to their method of preservation, juices are divided into fresh-squeezed, chilled, frozen, pasteurized, and concentrated [6,7]. Currently, consumers prefer to opt for fruit juices as an easy way to consume the five servings of fruits and vegetables recommended by the World Health Organization (WHO) for a healthy diet [8]. The low pH value of fruit juices reduces the growth of pathogenic microorganisms, making these products safe and attractive to consumers [9].

In recent years, many cases related to microbiologically spoiled fruit juices, leading to heavy economic losses for industry, were reported. These spoilages were mainly related to the flavor-producing bacteria, which impoverished the sensory characteristics of the products and made them unacceptable to consumers [10,11,12]. The main source of microorganisms in this type of food is the environment, primarily through contaminated soil, water, process machinery, and filling lines with inadequate hygiene protocols [13,14,15]. The microorganisms found on the surfaces of fruits and vegetables are an abundant and diverse group [6,16]. This group includes bacteria, as well as yeasts and molds, and there is a clear relationship between the composition of the microflora and the chemical composition of fruits and vegetables, especially the ratio of carbohydrates to proteins and acidity [17,18]. In the juicing industry, the high hydrostatic pressure (HHP) technique effectively eliminates both saprophytic and pathogenic bacteria [19,20,21]. However, the challenge lies with spore-forming bacteria, which survive at high temperatures and during pressurization, and then germinate and grow in the food matrix [22,23]. The most widespread species responsible for juice spoilage is *A. acidoterrestris* [20,24]. The bacteria from the *Alicyclobacillus* genus include Gram-positive, aerobic, and spore-forming bacilli [25,26]. The contamination of the product with this bacterium is difficult to detect because there is no gas production or packaging swelling [26]. A noticeable sign of spoilage is a taste described as medicinal, phenolic, and antiseptic, which is mainly associated with the production of guaiacol (2-methoxyphenol) [27,28], but also halophenols, such us 2,6-dibromophenol, and 2,6-dichlorophenol [25,29].

The use of specific, obligately lytic bacteriophages is one of the methods that is increasingly considered to eradicate bacteria from environments in the food industry [30,31]. Phage cocktails can be used to disinfect production surfaces, during raw material washing, or they can be added directly to the finished product [31,32,33]. This biological method of food preservation has been approved for use in several countries outside the European Union, such as the USA, Canada, and Switzerland [33,34,35]. Recently, the use of bacteriophages in the food industry and therapeutics has become highly popular, and many scientific centers around the world research the development of effective phage biopreparations [36,37]. Bacteriophages can infect and replicate only in vegetative bacterial cells [38], making it difficult to use this method to eliminate spore-forming bacteria [39]. In an article by Butala and Dragoš [39], the relationship between phages and hosts that form spores was described. The authors found that phages specific to spore-forming bacteria may have evolved molecular mechanisms, enabling them to cope with various stages of host development. For example, prophages have eliminated the mechanisms of translocation of their genomes into the pre-spore. Instead, phage DNA is protected from environmental conditions until the spore germinates [36]. After spore germination, the continuation of the phage’s lytic cycle occurs. Prophages can alter the frequency of sporulation and modulate the timely induction of sporulation and spore germination [36]. Recently, it was shown that prophage phi3T targeting *Bacillus subtilis* increases the rate of sporulation and affects the spore-germination time [40].

The simultaneous application of high-pressure techniques and phage cocktails represents a potentially effective solution to the above problem. This combined method can reduce the growth of vegetative bacterial cells and increase the germination of spores, which can be subsequently eliminated by bacteriophages [41,42,43]. The necessary condition for selecting the appropriate phage is to determine its ability to infect the widest possible range of bacteria in the food environment and its multiplication cycle. Therefore, this study aimed at the genomic and functional characterization of a newly isolated bacteriophage targeting the guaiacol-producing strain of *A. acidoterrestris*, with an emphasis on its potential use in food biopreservation.

## 2. Materials and Methods 

### 2.1. Source of Bacterial Host Strains 

Forty-six bacterial strains used in this research were deposited in the Culture Collection of Industrial Microorganisms—Microbiological Resources Center of the Department of Microbiology, at the Prof. Wacław Dąbrowski Institute of Agricultural and Food Biotechnology—State Research Institute (IAFB; Warsaw, Poland). All strains were isolated from food products. The ability to produce guaiacol was determined for each of the strains [44,45]. The belonging of the strains to the *Alicyclobacillus* genus was confirmed by amplification of the *16S* rRNA gene region. Sequencing was outsourced to Genomed S.A. company (Warsaw, Poland). Raw sequences were analyzed using BLASTn (NCBI) and deposited in the GenBank database. Guaiacol-producing *A. acidoterrestris* strain KKP 3133 was isolated in 2002 from apple-juice concentrate, and deposited as TO-117/02 [23,41].

### 2.2. Bacteriophage Isolation

*Alicyclobacillus* phage strain KKP 3916 was isolated from soil sample originating in an apple orchard near Warsaw, Poland. In total, 200 mL of saline solution was poured over the 20 g of soil, which was left overnight on a shaker set at 180 rpm, 23 °C (Innova 4000 Incubator Shaker, New Brunswick Scientific Co., Inc. Edison, NJ, USA). Subsequently, the mixture with soil was centrifuged at 8000× *g* for 8 min (ultracentrifuge Sorvall LYNX 6000, Thermo Fisher Scientific, Watertown, MA, USA) to separate bacteria and soil from the solution with bacteriophages. The supernatant was filtered through a 0.45 µm syringe filter (Minisart^®^ NML Cellulose Acetate; Sartorius, Goettingen, Germany). Aiming at the isolation of phages targeting *Alicyclobacillus* sp., 20 mL of double-concentrated BAT broth (Merck, Darmstadt, Germany), 20 mL of filtered soil solution, and 1 mL of an overnight *Alicyclobacillus* culture were added to 50 mL falcons. The phage-culture medium was incubated at 45 °C for 24 h. Next, the culture was centrifuged at 8000× *g* for 10 min to separate bacteria from the proliferated bacteriophages. The supernatant was filtered through a 0.45 µm syringe filter [46]. The concentration of the phages (phage titer; in PFU mL^−1^) in the obtained lysates was determined using the double-layer agar-plate method in triplicate [47,48]. For this purpose, a series of dilutions of phage lysate was prepared. In total, 100 μL of fresh bacterial suspension in BAT broth was added to 500 μL of appropriately diluted lysate. The suspension was gently stirred and left at 20 °C for 20 min to allow the phages to adsorb to the host-cell surface. After adsorption, the suspension was poured onto a solid BAT agar (Merck, Germany) and 4 mL of liquid and cooled to 50 °C soft BAT agar (BAT broth with 0.75% agar-agar, pH 4.0–4.2) was added. The whole mixture was stirred and spread evenly over the surface of the solid medium. After drying, the dishes were kept upside-down overnight at 45 °C. 

### 2.3. Purification of Bacteriophages

In order to purify the isolated phages, the resulting individual plaque was cut with an injection needle and transferred to the tubes supplemented with 1 mL of SM buffer (5.8 g L^−1^ of sodium chloride, 2.0 g L^−1^ of magnesium sulfate seven-hydrate, 50 mL of 1 M Tris-HCl (pH 7.5), 5 mL of 2% gelatin solution), and 10 μL of chloroform [49]. The tubes were left at 20 °C on a laboratory cradle for 2 h to diffuse the viral particles from the agar into the buffer and to inactivate the remaining bacteria in the lysate. One mL of fresh bacterial culture and the resulting purified phage suspension were transferred into tubes containing 40 mL of BAT broth. After incubation overnight at 45 °C, the suspension was centrifuged at 8000× *g* for 10 min and filtered through a 0.45 μm filter. In the lysate obtained in this way, phage titers were determined using the double-layer agar-plate method [47,48]. Purification was performed in four rounds of single-plaque passage. The resulting lysates were stored in refrigerator (at 4 °C) and frozen with the addition of 20% glycerol at −80 °C and −150 °C.

### 2.4. Determination of the Bacterial Hosts Range

In this study, fifty-three *Alicyclobacillus* strains were used to examine the range of bacterial hosts of the bacteriophage (of which forty-six strains were from the IAFB, and seven strains were from the Leibniz Institute DSMZ-German Collection of Microorganisms and Cell Cultures). Purified and amplified phage lysates (phage titers ~10^6^ PFU mL^−1^) were used and the double-layer agar-plate method was applied to determine the phage’s ability to infect selected bacteria strains [47,48]. Plates were incubated at 45 °C for 24 h in triplicate. After incubation, clearing zones were determined on any bacterial digests recorded according to the assays: “++”—transparent plaques; “+”—cloudy plaques; “−”—no plaques (insensitive bacterial strain).

### 2.5. Changes in the Growth Kinetics of A. acidoterrestris Strain KKP 3133 after Phage Infection

The kinetic growth of *A. acidoterrestris* strain KKP 3133 infected with newly isolated bacteriophages was measured in ten replicates using a Bioscreen C Pro automated growth analyzer (Yo AB Ltd., Growth Curves, Helsinki, Finland). A growth curve was prepared for bacterial strain (data unpublished in this paper). For this purpose, the bacterial culture was inoculated every hour onto BAT agar, incubated at 45 °C for 24 h, and optical density was measured simultaneously (DU^®^ 640 Spectrophotometer, Beckman Instruments, Inc., Fullerton, CA, USA). The dependence of optical density on the number of bacterial cells was determined (performed in triplicate). 

The overnight bacterial culture was diluted with BAT broth until the optical density was appropriate for the phage titer. Phage lysates were prepared so that the values of MOI (multiplicity of infections) were 1.0. and 0.1. Next, 180 µL of BAT broth was pipetted into multi-well plates and 10 µL of appropriately diluted bacterial culture and phage lysate was added. The plate was placed in the Bioscreen C Pro for 24 h with an average stirring intensity of 15 s before measurement. Optical density measurement was performed every 30 min using a wide band of wavelengths, ranging from 400 nm to 600 nm (OD_400–600_). The effectiveness of bacterial growth inhibition resulting from phage application was assessed with regard to a control culture that contained only bacteria. Based on the obtained curves illustrating the dependence of the change in optical density and the duration of the culture, the specific growth-rate coefficient (µ) was calculated according to the following formula:µ = (ln OD_max_ − ln OD_min_)/Δt(1)
where ln OD_max_—is the natural logarithm of the maximum value of the exponential growth of the culture, ln OD_min_—is the natural logarithm of the minimal value of exponential growth of the culture, and Δt—duration of the exponential growth of culture, (h).

### 2.6. Effect of Selected Factors on the Preservation of the Phage Activity

In this stage of the experiment, the stability of bacteriophages after exposure to a wide range of temperatures, pH values, and UV radiation times was examined. To determine the activity of phage lysates at various levels of active acidity, 100 μL of phage lysate was added to test tubes containing 9.9 mL of sterile saline (0.85% NaCl) with a fixed active acidity (pH 2–12). The mixture was maintained at 20 °C for 1 h. To determine phage activity at different temperatures (−20 °C, 4 °C, 20 °C, 30 °C, 40 °C, 50 °C, 60 °C, 70 °C, and 80 °C), 100 μL of the phage lysate was added to 9.9 mL of saline with pH 7.0. The mixture was maintained for 1 h at the specified temperatures. To determine the effect of UV radiation, phage lysate was exposed to UV for 0, 5, 10, 15, 25, 30, and 60 min. The experiments were carried out in three independent replicates. After incubation under the selected physical or chemical conditions, the activity of phages was determined through the double-layer agar-plate method.

### 2.7. Determination of Morphological Characteristics of Phage

Transmission-electron microscopy (TEM) was used to determine the morphological characteristics of the isolated bacteriophage. Phage lysate in a volume of 1 mL was centrifuged at 20 °C at 14,500 rpm for 40 min (MiniSpin^®^ plus centrifuge, Eppendorf, Hamburg, Germany). The supernatant was removed, and the precipitate was suspended in 2 mL of 100 mM cold ammonium acetate (filtered through a 0.22 μm syringe filter). The precipitate was dissociated by repeated pipetting and centrifuged again. The whole procedure was repeated four times. After centrifugation, the precipitate was rinsed off the wall of the Eppendorf tube with 50 μL of ammonium acetate according to the procedure of Ackermann [49], with modification. Two μL of phage suspension in ammonium acetate was coated onto carbon-sputtered copper–wolfram mesh grids. After drying, the preparation was stained for 1 min in a 2% uranyl acetate solution (Warchem, Warsaw, Poland). Prepared samples were dried for 12 h at ambient temperature under sterile conditions [50,51] and visualized under JEM 1400 PLUS transmission-electron microscope (Japan Electron Optics Laboratory Co., Ltd., Tokyo, Japan) at 100,000–200,000× magnification, at a voltage of 80 kV [52].

### 2.8. Extraction of Bacteriophage Genomic DNA

Bacteriophage genomic DNA from lysates was isolated using the PureLink^TM^ RNA/DNA Mini Kit (Thermo Fisher Scientific Inc., Carlsbad, CA, USA) according to the manufacturer’s protocol, with modifications by Wójcicki et al. [53]. The first step was to concentrate the phage lysate by ultracentrifugation. For this purpose, 40 mL of phage lysate and 8 mL of precipitating solution (PEG-NaCl: 2.5 M NaCl, 20% PEG 8000) were transferred into bottle for ultracentrifugation. The resulting solution was incubated overnight on ice. The lysate was then centrifuged at 27,000 rpm for 1.5 h at 4 °C (Sorvall LYNX 6000 ultracentrifuge, Thermo Fisher Scientific, Watertown, MA, USA). The supernatant was carefully removed, and the phage pellet was resuspended in 400 µL Lysis Buffer (containing 5.6 µg Carrier RNA) and vortexed. Subsequently, 50 µL of the proteinase K was added and the prepared solution was incubated for 1 h at 56 °C with shaking at 900 rpm (ThermoMixer C, Eppendorf, Hamburg, Germany). After incubation, the tube was briefly centrifuged, and 300 µL of 100% ice-cold ethanol (molecular biology grade) was added. The tube was vortexed for 10 s and left at room temperature for 5 min. After incubation, sample was briefly vortexed to remove droplets from the tube walls. The content of the tube was transferred to a viral spin and centrifuged for 1 min at 10,000 rpm (MiniSpin^®^ plus centrifuge, Eppendorf, Hamburg, Germany). Next, 500 µL of wash buffer was added and centrifugation for 1 min at 10,000 rpm was carried out. The filtrate was removed, and the column was centrifuged for 3 min at 14,500 rpm. The next step was the elution of the genetic material. The viral spin column was transferred to a new tube and 20 µL of RNase-free water was added. The sample was incubated for 1 min at 20 °C and then centrifuged for 1 min at 14,500 rpm. The filtrate was applied to the same column and centrifuged again. The purity of the obtained DNA was evaluated with a Nanodrop ND-1000 spectrophotometer (Thermo Fisher Scientific, Watertown, MA, USA), and DNA concentration was determined with a Qubit 4.0 fluorometer using the Qubit dsDNA BR Assay Kit (Invitrogen, Carlsbad, CA, USA). The DNA was stored at 4 °C until whole-genome sequencing (WGS) analysis.

### 2.9. Genome Sequencing and Bioinformatics Analysis

The company genXone SA (Złotniki, Poland) was commissioned to sequence the entire phage genome. The DNA library was prepared using Rapid Barcoding Kit reagents (Oxford Nanopore Technologies, Oxford, UK), according to the manufacturer’s protocol. A sequencing depth of at least 50× genome coverage was assumed. The NGS sequencing was performed using the nanopore technology on the GridION X5 sequencing device (Oxford Nanopore Technologies, Oxford, UK) under the control of MinKnow v22.10.5. Guppy v6.3.8 (Oxford Nanopore Technologies, Oxford, UK), which was applied to call bases and to perform barcode demultiplexing, generating a .fastq file for each barcode. The de novo assembly of genome was performed in Flye v2.8.1 software [54], while annotation of phage genomes was performed in Phanotate v1.5.0 [55] and PhaGAA [56] software. Proksee [57] software was used to visualize phage genome. The viral proteomic tree of phage genome was calculated by BIONJ based on genomic distance matrixes and the mid-point rooted, and was represented in the circular view. Branch length was log-scaled. The sequence and taxonomic data were based on Virus-Host DB [58]. The tree was generated using the ViPTree server [59]. Phage similarity on scatter 2D plot computationally was predicted and rendered through PhageAI software [60]. The phage genome was deposited in the GenBank database.

### 2.10. Statistical Analysis

All experiments were repeated at least three times. Data presented graphically or in tables were subjected to statistical analysis using Graph Prism 8.02 software (GraphPad Software Inc., San Diego, CA, USA). One-way ANOVA followed by Tukey’s test with a 95% confidence interval (α = 0.05) was used to evaluate the effect of selected physical and chemical factors on phage activity.

## 3. Results and Discussion 

### 3.1. Determination of Plaque and TEM Morphology of Bacteriophage

Plaques usually result from a combination of phage propagation, viral diffusion, and the lysis of bacterial hosts. Plaques can also result from partial bacterial growth suppression in connection with phage propagation, even without direct bactericidal action [61,62]. The morphology of phage plaques is important for initial identification, as transparent plaques indicate lytic (virulent) bacteriophages and cloudy plaques indicate lysogenic (temperate) bacteriophages [62]. The size of the bacteriophage is inversely proportional to the size of the plaque, as a small bacteriophage diffuses more easily in soft agar, causing a large plaque diameter [63,64]. In addition, the morphologies of lysates are linked to such fitness characteristics as the size of the burst, the adsorption rate and lysis time, and the diffusion rate of the phage in a particular medium [65]. It is also possible to identify phage mutants based on changes in plaque morphology compared to the characterized wild-type phage, including molecularly engineered phages [66,67].

Figure 1A shows the plaque obtained for the *Alicyclobacillus* phage strain KKP 3916. The plaques were characterized by a transparent center with cloudy edges. Electronograms obtained using TEM allowed the visualization of the morphology of the bacteriophage (Figure 1B,C). The bacteriophage has a complex structure (tailed phages), containing an icosahedral symmetrical head (capsid) and a long, contractile tail.

### 3.2. Range of Bacterial Hosts

Forty-six strains isolated from food products were used to define the bacterial host range (Table 1). Moreover, in our study, we used seven strains from the Leibniz Institute DSMZ-German Collection of Microorganisms and Cell Cultures isolated from different environments (Table 1). 

The *Alicyclobacillus* phage strain KKP 3916 infected 22.6% (12/53) of the tested strains from the *Alicyclobacillus* genus. Most of the phage lysis zones observed were transparent plaques with cloudy edges. Examples of plaques on BAT agar are shown in Figure 2. 

Of the 41 strains of the *A. acidoterrestris* species, the newly isolated bacteriophage had the ability to infect nine. No lysis zone was observed against the strains of *A. hesperidum, A. herbarius,* or *A. acidophilus.* Out of four strains from *A. fastidiosus* species and five strains from *A. acidocaldarius* species, the *Alicyclobacillus* phage strain KKP 3916 had the ability to infect one (KKP 3002) and two (KKP 3135 and KKP 3157) strains, respectively. Of the twelve bacterial strains infected by *Alicyclobacillus* phage strain KKP 3916, nine were guaiacol-producing.

The host range plays an important role in the use of bacteriophage in therapy or food biocontrol. The elimination of only specific bacterial strains requires a narrower host range [68,69], to prevent the other microorganisms from being compromised (which is important in some industries, e.g., dairy). To eliminate pathogenic bacteria within a species, it is desirable to search for bacteriophages with wide host ranges [70]. This is analogous to the use of broad-spectrum antibiotics without the need to identify the pathogen or sensitivity to the antibiotic. In the absence of broad-spectrum phage isolation, a preparation containing several bacteriophages can be used [68,70,71].

### 3.3. Evaluation of the Activity of Phage against Bacterial Hosts

The effectiveness of a phage in inactivating bacteria is one of the critical characteristics that should be considered for potential candidates in phage therapy or food biocontrol. To evaluate this feature in relation to the tested bacteriophage, growth curves were plotted for each bacterial host strain using the Bioscreen C Pro automated growth analyzer. The growth curves showed that the *Alicyclobacillus* phage strain KKP 3916 was highly effective against the *A. acidoterrestris* strain KKP 3133 at both tested MOIs (Figure 3).

Moreover, in these samples, there were no differences in inhibition effect during the first 6 h after infection and after 20 h of incubation. The addition of phages to the culture of the *A. acidoterrestris* strain KKP 3133 caused a shorter logarithmic phase of host growth compared to the control culture. A slight increase in optical density may indicate the acquisition of resistance to the phage used, a transition to the lysogenic cycle, or an increase in the number of cells [72,73]. In the course of their evolution, bacteria have developed a number of defense mechanisms to protect cells from phage infection. The primary defense mechanism of bacteria is the physical masking of the receptor through changes in its conformation or its complete disappearance from the cell envelope. The physical masking of receptors involves the production of an additional envelope that limits the interaction of the bacteriophage with the bacterial cell [74,75]. Genetic mutations result in the development of mutants that are completely insensitive to phage adsorption [76]. Often, once a host cell is infected with a phage, resistant bacteria can destroy the foreign genome without reducing their own viability or damaging their own genetic material. In this case, the phage DNA is degraded in the cytoplasm as a result of endonucleolytic digestion. A key role in this process is played by the restriction/modification system. This system consists of restrictionases that recognize and cut specific DNA sequences and modify them by adding methyl groups [75,77]. Cells with this restriction system have their chromosomal DNA modified by a methyltransferase enzyme that adds specific sequences. This protects against the insertion of cuts into the bacterial DNA. The phage DNA does not have a methylation pattern, so it is not degraded by restriction enzymes once it enters the cytoplasm [78]. A key role in the defense mechanisms of prokaryotic organisms is played by the CRISPR/Cas system. This is a sequence of clustered, regularly spaced, short palindromic sequences, which detects and destroys phage DNA [79]. This provides a type of immune-defense system in bacteria and archaeons. The system works by “remembering” a specific nucleic acid sequence inserted into the cytosol. During phage infection, the initial step in the response the recognition of phage DNA and its cutting by Cas proteins [80].

The changes in the optical density of the tested bacterial strains after the incubation with the phage are shown in Table 2. The lower values of the specific growth-rate coefficient (μ) determined for the phage-infected cultures indicate the significant inhibition of cell divisions in the log phase.

Similar results were obtained in the study conducted by Kozantsev et al. [73], where the growth kinetics of *Bacillus cereus* strain VKM B-370 after infection with the lysogenic phage B13 at different MOIs was determined. It was shown that the higher the MOI, the faster the partial growth inhibition was achieved. It is noteworthy that the difficulty in eliminating spore-forming bacteria lies not only in the resistance of bacterial spores to factors such as UV, heating, low pH, and pressure, but also in phage infection. This has also been confirmed in studies on the elimination of *B. subtilis* and *B. cereus* using the phages PBSC1 and PBSC2 [81]. The use of phages caused a significant inhibition of bacterial culture growth, but due to the development of phage-resistant bacterial spores, the optical density increased at the end of the culture. The use of phage biocontrol combined with other antimicrobial agents to eliminate bacteria may be an effective solution [81,82,83].

### 3.4. Effect of Environmental Conditions on the Phage Activity

The determination of the stability of phages helps to predict how bacteriophages may behave in the environment. In this study, the activity of isolated phages exposed to a wide range of temperatures (from −20 to 80 °C), levels of active acidity (pH from 2 to 12), and times of UV exposure (0, 5, 10, 15, 20, 25, 30, and 60 min) was evaluated. The activity was expressed as phage titer (in log PFU mL^−1^). The effect of temperature on the activity of the phages is presented in Figure 4.

The *Alicyclobacillus* phage strain KKP 3916 retained its activity at both low and high temperatures. Temperatures from 4 °C to 30 °C slightly affected the activity of the bacteriophages, while freezing temperatures (−20 °C) reduced their activity by about 99% in relation to the control culture (20 °C). Temperature increases to 40 °C, 50 °C, and 60 °C caused a denaturation of the phage virions due to stress conditions and, consequently, reduced their activity by about 99% compared to the control. The incubation at 70 °C resulted in the almost complete inactivation of the bacteriophages (99.9%), whereas the application of 80 °C resulted in PFU reduction to undetectable levels, probably due to the denaturation of the bacteriophage protein.

In most cases, bacteriophages specific to thermophilic bacteria are resistant to high temperatures [62]. In a study by Morozov et al. [84], the thermophilic bacterium *Aeribacillus* sp. was the bacterial host of the tested AP45 phage. The optimal temperature for the growth of this bacterium is 55 °C, while the optimal temperature for maintaining the specific phage viability was below 75 °C. The AP45 bacteriophage showed stability after incubation for 24 h at 85 °C and 1 h at 95 °C. Another example is the effect of temperature on phage activity against *Campylobacter jejuni* [85]. The incubation at 42 °C had the best effect on the phage stability, while temperatures of 55 °C and 4 °C reduced its activity by about 99.9%.

The effect of a wide range of levels of active acidity on the activity of the phages is presented in Figure 5. The isolated bacteriophage had a high tolerance to active acidity. No statistically significant differences were observed after the incubation of the lysate in solutions with pH ranging from 3–11. An alkaline environment (pH = 12) reduced the activity of the phage by 99.9%. The solution at pH = 2 caused the complete denaturation of the phage virions. 

In the study by Capra et al. [86], it was shown that the effect of pH on the activity of *Lacticaseibacillus casei* and *Lacticaseibacillus paracasei* (former names *Lactobacillus casei* and *Lactobacillus paracasei*) phages was dependent on the type of phage. Most of the bacteriophages showed high activity after incubation for 30 min in the pH range between 4 and 11. Similar to the results obtained in our study, the incubation at pH 2 resulted in the complete elimination of bacteriophages due to the denaturation of the virion head [86]. In other studies, *B. subtilis* bacteriophages showed activity in the pH range of 6.0–8.0 after incubation for 60 min. Incubation at pH extremes of 2–3 and 10–11 often resulted in a complete loss of infectivity [87].

Figure 6 shows the effect of UV-exposure time on the phage activity. Irradiation for 5 min resulted in an insignificant decrease in phage stability; however, a 10-minute exposure to UV resulted in a reduction in its activity by almost 99% compared to the control sample. A phage reduction of almost 99.9% was observed after exposure to UV for 15 min and 20 min. In contrast, exposures for 25 min and 30 min reduced the number of phages by 99.99% compared to the control. No plaques were observed after 60 min of UV exposure.

Research data concerning the stability of bacteriophages subjected to UV exposure are scarce. According to Hazem [88], thermophilic *Bacillus*-specific bacteriophages 46 and 50 were stable after 13 min and 20 min of UV exposure, respectively, while the lytic bacteriophage against *Salmonella* SS3e lost 50% viability after 1 min, 90% of phage particles were inactivated after 5 min, and no phage particles were detected after 15 min of UV exposure [89].

In addition to environmental factors, the presence of metal salts has a significant impact on the adsorption process during phage–bacterial-host interactions. Numerous studies indicate that most divalent metal cations can increase the efficiency of bacterial lysis and, thus, the invasiveness of bacteriophages [90]. Divalent metal cations are believed to enhance the structures of bacterial viruses, such as phage T4 [90]. The formation of a complex between the phage structure and metal ion inhibits the inactivation of the phage [91]. Metal sorption on the surfaces of viruses can not only contribute to nanoparticle-metal transport, but also increase phage infectivity. In a study by Carey-Smith et al. [92], T5 bacteriophages were examined, and the obtained results indicated that viruses are inactivated much more rapidly by heat when suspended in 0.1 M sodium salt solutions than in broth. The inactivation rate of the T5 phage in 0.1 N NaCl at 37 °C can be significantly reduced by adding 10^−8^ M of the following divalent cations: Ca, Mg, Ba, Sr, Mn, Co, Ni, Zn, Cd, and Cu. Increases in the concentration of cations in the suspending medium increase the resistance of the T5 phage to the inactivating effects of temperature, and the stability of virions in the presence of various cations results from the formation of a phage complex with metal ions. The results obtained by Chhibber et al. [93] indicate that the MR-10 MRSA phage required the presence of calcium ions for primary attachment to bacterial cells (*Staphylococcus aureus* strain 43300). It is likely that calcium is used to introduce the phage’s genetic material into the cytoplasm of host cells.

### 3.5. Analysis of Phage Genome

The complete genome of the *Alicyclobacillus* phage strain KKP 3916 was sequenced and deposited in the GenBank database under accession number OQ846916. In addition, the phage was deposited in the Culture Collection of Industrial Microorganisms—Microbiological Resources Center of the Department of Microbiology, at the Prof. Wacław Dąbrowski Institute of Agricultural and Food Biotechnology—State Research Institute. 

As with the TEM, the genome analysis confirmed that the phage belongs to the *Caudoviricetes* class. Figure 7 shows the proteomic tree generated using the ViPTree. It indicates that the isolated bacteriophage is a distant relative of phages from the *Herelleviridae* family. 

The sequence of the *Alicyclobacillus* phage strain KKP 3916 genome in the form of linear dsDNA is 120,131 bp long with 40.3% G+C pair content. Out of the 204 predicted open reading frames (ORFs), 70 ORFs are associated with genes encoding proteins with known functions and 134 ORFs encode hypothetical proteins with unknown functions (Figure 8).

The annotated functional proteins were divided into several groups, depending on their function: activities related to metabolism and replication, genome packaging, structure, lysis, and lysogeny. The main group of proteins whose function was predicted comprised those related to metabolism and replication. Among genes related to replication and metabolism, the following proteins were found in the phage genome: DNA helicase (DnaB-like replicative helicase), DNA primase, RNA ligase, DNA polymerase, DNA methyltransferase, transposase, exonuclease, and other proteins (Figure 7 and Appendix A). The group of other proteins included those associated with phage structure, including: portal protein, capsid, tail, and lipoprotein. The proteins associated with lysis include, among others, holin, endolysin, and spanin. These enzymes are capable of disrupting bacterial peptidoglycan from within, leading to bacterial lysis and the release of new phages [94,95]. Small hydrophobic proteins, called holins, accumulate in the inner cell membrane, oligomerize, and form pores in the membrane, causing the activation of endolysin, which accumulates in the cytoplasm or periplasm [94,95,96]. Endolysins, known as peptidoglycan hydrolases, are a class of enzymes encoded by genes of bacteriophages that degrade the bacterial cell wall at the end of the lytic cycle [94,95]. In Gram-negative hosts, there is often a third protein whose action is required for complete cell lysis, namely spanin, which breaks down the last barrier—the outer membrane [97,98]. Spanins are lysing proteins, essential for disrupting the outer membrane of the bacterial host at the final stage of bacterial lysis [96,99]. Most phages produce a two-component spanin complex, consisting of an outer-membrane lipoprotein (o-spanin) and an inner-membrane protein (i-spanin) with a periplasmic domain [94,99,100]. In the *Alicyclobacillus* phage strain KKP 3916, one region encoding Rz-like spanin, two regions encoding holin, and three regions encoding endolysin were predicted. The analysis of the *Alicyclobacillus* phage strain KKP 3916 genome showed five tRNAs. The presence of tRNA, especially in virulent phage genomes, is a common phenomenon. Canchaya et al. [101] suggested that the presence of tRNA in the phage genome can be correlated with the better integration of virulent phages with the bacterial host chromosome. Bailly-Bechet et al. [102] reported that there is a positive association between the size of the phage genome and the number of tRNA genes it contains. No genes related to antibiotic resistance were found in the genome of the newly isolated phage, but four integration-related regions were identified. Integrase integrates the phage’s nucleic acid into the bacterial host genome, which does not directly lead to bacterial destruction, but indicates a moderate (lysogenic) bacteriophage life cycle [103]. The integrated or extra-chromosomal form of the phage is called a prophage, while a bacterial cell with an integrated prophage genome is called a lysogen. External stressors (especially those that pose a threat to the bacterial host cell) can initiate the transition from the lysogenic to the lytic cycle [104,105]. In addition, other regions associated with lysogeny were noted in the *Alicyclobacillus* phage strain KKP 3916 genome: the toxin gene and excisionase. Bacteriophages encoding virulence factors can convert their bacterial hosts through a process known as the lysogenic conversion of the phage from a non-pathogenic strain to a virulent strain or a strain with enhanced virulence. An example of the provision of bacteria with a toxin gene is a group of E. coli strains that produce Shiga toxins [106]. It has been suggested that prophages increase the ecological adaptation of many bacteria, regardless of their characteristics. Prophages can, among other activities, modulate bacterial resistance to various environmental factors (such as low pH), increase antibiotic tolerance, or facilitate biofilm formation [102,107,108]. It is also known that lysogeny requires the expression of bacteriophage integrase, while the excision of prophage DNA from the host genome requires an additional phage-encoded protein, called excisionase. Although this process is often accompanied by the lysis of host cells, the curing of prophages from host bacteria has been reported in some cases. It is worth noting that integrase and excisionase proteins are considered markers of lysogenic infection [109].

Based on the morphology and the comparison of its protein regions, *Alicyclobacillus* phage strain KKP 3916 was assigned to viruses with complex structures (tailed phages from the Caudoviricetes class). However, analyses of the phylogenetic relationship prevented its unambiguous assignment to a specific family and genus. The weak similarity with other phage genomes deposited in the databases suggests that the isolated bacteriophage may be representative of a new genus of tailed bacteriophages.

## 4. Conclusions

The application of bacteriophages in food biocontrol is gaining increasing interest among scientists around the world, including the European Union. Therefore, in this study, we isolated a phage active against guaiacol-producing *A. acidoterrestris* and investigated its biological properties. To the best of our knowledge, this is the first study on the isolation and characterization of a phage specific to bacteria from the *Alicyclobacillus* genus. Arguably, a new genus of tailed bacteriophages was discovered. It was shown that the newly isolated phage exhibits a narrow host spectrum and broad tolerance to environmental factors, such as temperature, pH, and UV radiation. The bioinformatics analyses demonstrated that the genome of the *Alicyclobacillus* phage strain KKP 3916 contains sequences corresponding to integration into the host genome, suggesting that this virus is not strictly lytic. Due to the risk of its potential involvement in horizontal gene transfer, it is not an appropriate candidate for food biocontrol. Further research will address the characterization of newly isolated bacteriophages against spore-forming bacteria and their effectiveness in eliminating food-spoilage-causing strains either individually or combined with high pressure. The effectiveness of bacteriophages will be examined in plant-based food matrices.

## Figures and Tables

**Figure 1 genes-14-01303-f001:**
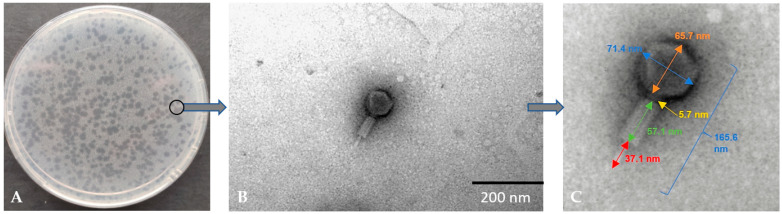
Morphology of plaques on double-layer agar plate (**A**) and electron micrographs from the TEM (**B**,**C**) of *Alicyclobacillus* phage strain KKP 3916 targeting *A. acidoterrestris* strain KKP 3133.

**Figure 2 genes-14-01303-f002:**
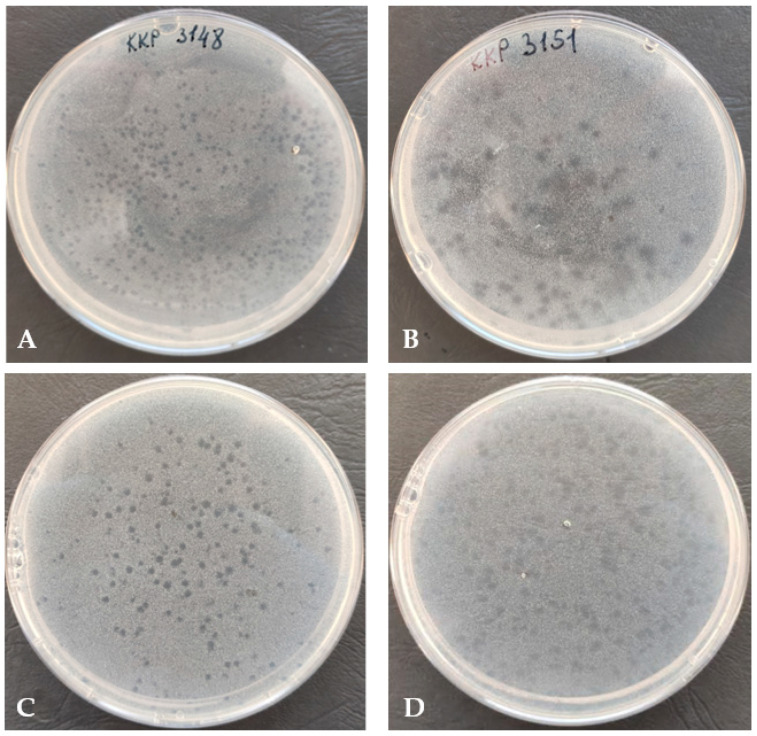
Plaque morphology (on double-layer agar plate) of *Alicyclobacillus* phage strain KKP 3916 infecting *A. acidoterrestris* strain KKP 3148 (**A**), *A. acidoterrestris* strain KKP 3151 (**B**), *A. acidocaldarius* strain KKP 3157 (**C**), and *A. acidoterrestris* strain KKP 3195 (**D**).

**Figure 3 genes-14-01303-f003:**
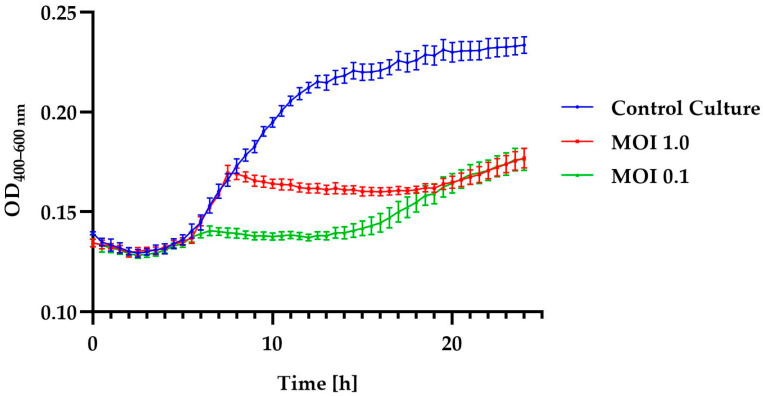
Growth curves of *A. acidoterrestris* strain KKP 3133 treated with *Alicyclobacillus* phage strain KKP 3916 at infection coefficients of MOI = 1.0 (red line) and MOI = 0.1 (green line) compared to the control culture (blue line). Points represent the mean (*n* = 10); error bars represent the standard deviation (± SD) of the optical density.

**Figure 4 genes-14-01303-f004:**
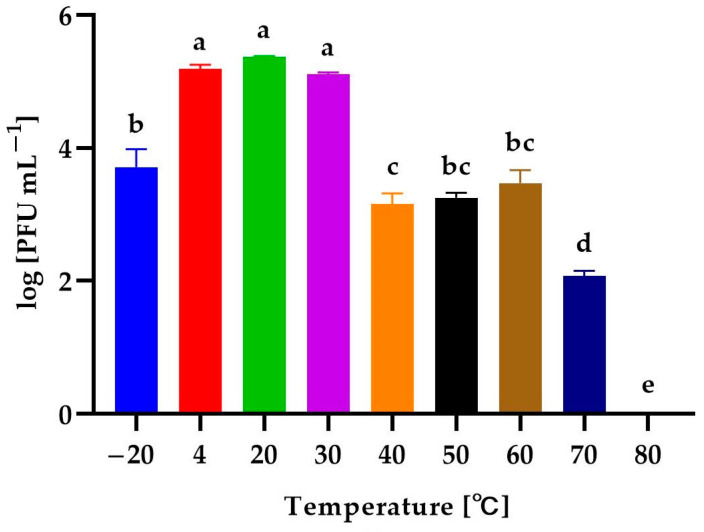
The activity of *Alicyclobacillus* phage strain KKP 3916 against *A. acidoterrestris* strain KKP 3133 after exposure to a wide range of temperatures. Letters a, b, c, d, and e indicate homogenous groups at a significance level of *p* ≤ 0.05, *n* = 3.

**Figure 5 genes-14-01303-f005:**
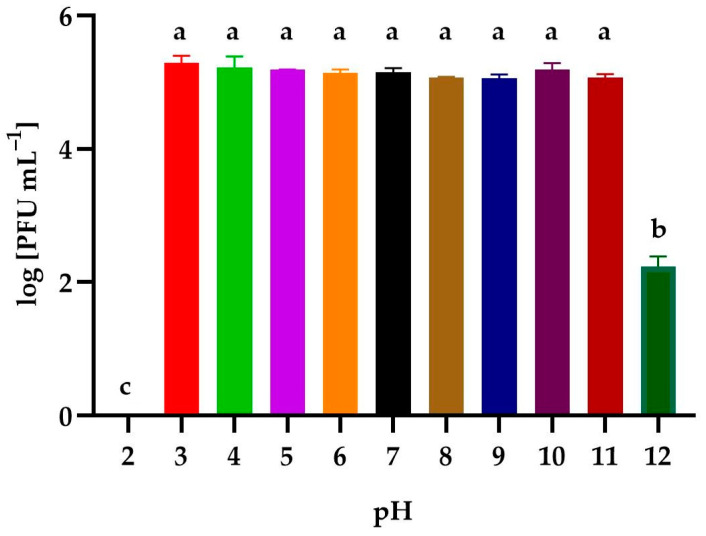
The activity of *Alicyclobacillus* phage strain KKP 3916 against *A. acidoterrestris* strain KKP 3133 after exposition to a wide range of pH values. Letters a, b, and c indicate homogenous groups at a significance level of *p* ≤ 0.05, *n* = 3.

**Figure 6 genes-14-01303-f006:**
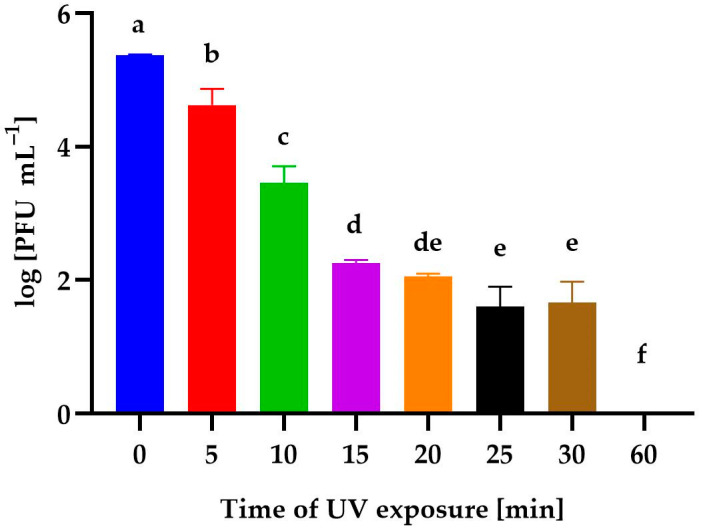
The activity of *Alicyclobacillus* phage strain KKP 3916 against *A. acidoterrestris* strain KKP 3133 after different UV-exposure times. Letters a, b, c, d, e, and f indicate homogenous groups at a significance level of *p* ≤ 0.05, *n* = 3.

**Figure 7 genes-14-01303-f007:**
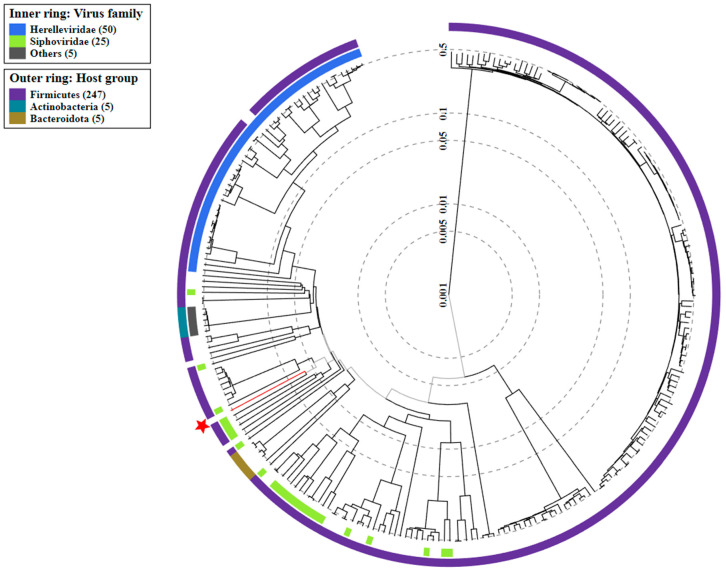
The viral proteomic tree of the *Alicyclobacillus* phage strain KKP 3916 is shown in a circular view. The branch represented by the phage under study is marked with an asterisk. The colored rings indicate the virus family (inner rings) and host groups (at the phylum level; outer rings). The tree was calculated by BIONJ based on the genomic distance matrix and rooted at the midpoint. Branch lengths are logoscaled. Sequence and taxonomic data were based on the Virus-Host DB [58]. The trees shown were generated using the ViPTree server [59].

**Figure 8 genes-14-01303-f008:**
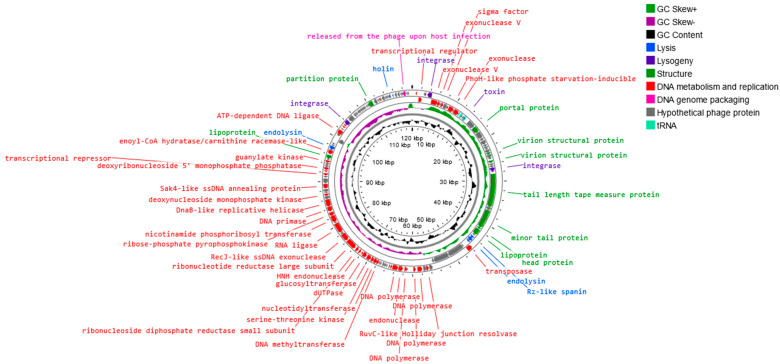
Map of the genomic organization of *Alicyclobacillus* phage strain KKP 3916 generated using the Proksee program [57].

**Table 1 genes-14-01303-t001:** Range of bacterial hosts of the genus *Alicyclobacillus* for the newly isolated *Alicyclobacillus* phage strain KKP 3916.

Bacterial Host Strain (*n* = 53)	Year of Isolation	Source of Isolation	Guaiacol Production	GenBank Accession Number	Infect with Phage
**Strains from the Culture Collection of Industrial Microorganisms—Microbiological Resources Center, IAFB (*n* = 46)**					
*A. acidoterrestris* strain KKP 377	2011	Concentrated cherry juice	+	OQ285871	++
*A. acidoterrestris* strain KKP 381	2005	Concentrated apple juice	+	OQ284561	−
*A. acidoterrestris* strain KKP 382	2005	Concentrated apple juice	+	OQ284562	−
*A. acidoterrestris* strain KKP 383	2006	Concentrated apple juice	+	OQ285847	−
*A. acidoterrestris* strain KKP 394	2005	Concentrated apple juice	+	OQ284103	−
*A. acidoterrestris* strain KKP 395	2002	Concentrated apple juice	+	OQ284104	+
*A. acidoterrestris* strain KKP 400	2002	Concentrated apple juice	+	OQ285872	−
*A. acidoterrestris* strain KKP 404	2012	Concentrated black currant juice	+	OQ284075	−
*A. acidoterrestris* strain KKP 564	2006	Concentrated apple juice	+	OQ285856	−
*A. acidoterrestris* strain KKP 596	2006	Concentrated apple juice	+	OQ285855	−
*A. acidoterrestris* strain KKP 609	2013	Concentrated strawberry juice	+	OQ285862	−
*A. acidoterrestris* strain KKP 610	2006	Concentrated apple juice	+	OQ285860	−
*A. acidoterrestris* strain KKP 611	2006	Concentrated apple juice	−	OQ285857	−
*A. acidoterrestris* strain KKP 613	2006	Concentrated apple juice	+	OQ318516	+
*Alicyclobacillus fastidiosus * strain KKP 3000	2019	Soil from a pear orchard	+	KY088044	−
*A. fastidiosus* strain KKP 3001	2019	Soil from a pear orchard	+	KY088045	−
*A. fastidiosus* strain KKP 3002	2002	Soil from a pear orchard	+	KY088046	+
*A. acidoterrestris* strain KKP 3133	2002	Concentrated apple juice	+	OQ261766	++
*A. acidoterrestris* strain KKP 3134	2004	Carbonated ingredient	+	OQ263350	−
*Alicyclobacillus acidocaldarius * strain KKP 3135	2004	Beverage powder	−	OQ285905	++
*A. acidoterrestris* strain KKP 3136	2006	Spoiled orange drink	+	OQ263355	−
*A. acidoterrestris* strain KKP 3137	2006	Concentrated apple juice	+	OQ263363	−
*A. acidoterrestris* strain KKP 3138	2006	Banana nectar	+	OQ262932	−
*A. acidoterrestris* strain KKP 3139	2007	Concentrated apple juice	+	OQ263354	−
*A. acidoterrestris* strain KKP 3140	2007	Concentrated apple juice	+	OQ262861	−
*A. acidoterrestris* strain KKP 3141	2008	Concentrated blackcurrant juice	+	KX371237	++
*A. acidoterrestris* strain KKP 3142	2008	Concentrated apple juice	+	KX371238	−
*A. acidoterrestris* strain KKP 3143	2008	Fresh apple	+	KX371239	−
*A. acidocaldarius* strain KKP 3144	2009	Sugar syrup	+	OQ285906	−
*A. acidoterrestris* strain KKP 3145	2009	Concentrated apple juice	+	KX371240	−
*A. acidoterrestris* strain KKP 3146	2009	Spoiled apple drink	+	OQ261717	−
*A. acidoterrestris* strain KKP 3147	2009	Concentrated apple juice	+	OQ263351	−
*A. acidoterrestris* strain KKP 3148	2009	Spoiled apple juice	−	OQ261718	++
*A. acidocaldarius* strain KKP 3149	2010	Concentrated strawberry juice	−	KX371241	−
*A. acidoterrestris* strain KKP 3150	2010	Concentrated red beet juice	+	KX371242	−
*A. acidoterrestris* strain KKP 3151	2011	Concentrated cherry juice	+	KX371243	++
*A. acidoterrestris* strain KKP 3152	2011	Concentrated cherry juice	+	KX371245	−
*A. acidoterrestris* strain KKP 3153	2011	Concentrated raspberry juice	+	KX371246	−
*A. acidoterrestris* strain KKP 3154	2011	Tomato juice	+	OQ285874	−
*A. acidoterrestris* strain KKP 3156	2012	Tomato juice	+	OQ263364	−
*A. acidocaldarius* strain KKP 3157	2013	Concentrated apple juice	−	OQ285907	++
*A. acidoterrestris* strain KKP 3194	2020	Soil from an apple orchard	+	KY088041	−
*A. acidoterrestris* strain KKP 3195	2020	Soil from an apple orchard	+	KY088042	+
*A. acidoterrestris* strain KKP 3347	2020	Soil from an apple orchard	+	KY088043	−
*A. acidoterrestris* strain KKP 3348	2020	Soil from an apple orchard	+	KY088047	−
*A. acidoterrestris* strain KKP 3349	2020	Soil from an apple orchard	+	MW332524	−
**Strains from the Leibniz Institute DSMZ-German Collection of Microorganisms and Cell Cultures (*n* = 7)**					
*Alicyclobacillus acidocaldarius* subsp. *acidocaldarius* strain DSM 446	1990	Acid hot spring	−	CP001727	−
*Alicyclobacillus hesperidum * strain DSM 12489	1998	Solfataric soils	+	FNOJ00000000	−
*Alicyclobacillus herbarius * strain DSM 13609	2000	Dry flower of *Hibiscus*	+	AUMH00000000	−
*Alicyclobacillus acidophilus * strain DSM 14558	2001	Acidic beverage, which was off-flavored	+	AB076660	−
*A. fastidiosus* strain DSM 17978	2003	Apple juice	−	NR_041471	−
*A. acidoterrestris* strain DSM 2498	1982	Juice	+	AB059675	++
*A. acidoterrestris* strain DSM 3922	1986	Garden soil	+	AURB00000000	−

Infected with phage: “++”—transparent plaques; “+”—cloudy plaques; “−”—no plaques (insensitive bacterial strain).

**Table 2 genes-14-01303-t002:** Changes in the optical densities of bacterial cultures after the addition of specific phages and values of the specific growth rate coefficient (μ).

	Control Culture	MOI = 1.0	MOI = 0.1
**ΔOD**	0.066	0.038	0.007
**μ [h** ** ^—1^ ** **]**	0.064	0.031	0.032

## Data Availability

Not applicable.

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
