# Peer review of "Characterization and Genome Study of a Newly Isolated Temperate Phage Belonging to a New Genus Targeting Alicyclobacillus acidoterrestris"

_genes, 2023, doi:10.3390/genes14061303_

Round 1
Reviewer 1 Report
Shymialevich explored the possibility of using (bacterio-)phages to combat Alicyclobacillus spp., a leading pathogen that spoils the juices and thus poses a significant problem for the juice industry. Inactivating Alicyclobacillus spp. is challenging due to its resistance to high temperatures and acidity. However, the use of phages shows promise as a solution. This study focused on isolating and characterizing a novel bacteriophage that targets Alicyclobacillus spp. The phage they isolated, Alicyclobacillus phage strain KKP 3916, was found to be active across a wide range of temperatures and pH values. However, it displayed reduced activity at high temperatures and upon exposure to UV radiation. The phage was classified as a tailed phage based on microscopic and genomic analyses. The genome sequencing revealed its characteristics, including the absence of antibiotic resistance genes but the presence of integration-associated regions, suggesting a temperate life cycle. This study is interesting because, to my knowledge, it is the first investigation into the isolation and genomic analysis of the phage specific to Alicyclobacillus spp. I am quite open to looking at a revised version if the authors could address some major and minor issues in a satisfactory fashion, which we describe in more detail below.
Major issues:
1. Some observations were mentioned without showing the results shown in the figures. For example, in lines 297-298, the authors mentioned that “Most of the phage lysis zones observed were transparent plaques with cloudy edges.”, but this observation is not obvious to me and perhaps other readers. I think it would be better if the authors could show the results directly in a figure.
2. Table 1: I believe the authors better give citations to the strains so that others can find those strains to reproduce their results. Alternatively, they can give references to the Industrial Microorganisms—microbiological resources center and Leibniz Institute DSMZ-German Collection of Microorganisms and Cell Cultures they mentioned. Additionally, I don’t understand the grey area near the bottom of the table. Are the GenBank Accession Numbers of those strains missing or not available? They better provide better clarification.
3. On lines 328-330, the authors mentioned the possibility of resistant bacteria. However, they don’t take a deep dive into the bacterial genome to figure out the potential mechanisms for the resistance. Although a genomic analysis may not give an answer, a discussion of possible resistant mechanisms is lacking. Thus, I would suggest they do a brief summary based on the following references: (1) restriction-modification (‪Geoffrey Wilson, Nucleic acids research 1991) and (2) CRISPR-Cas (Bridget N J Watson, et al, Cell Host Microbe, 2021; Shai Pilosof, SA Alcalá-Corona, et al, Nature Ecology and Evolution, 2020).
4. The authors investigated the influence of environmental factors such as pH and temperature. However, there are other important factors missing here such as metal concentrations. It has been shown in the past that Calcium concentration plays an important role in the adsorption of phages (Derek Ping, et al, ISME Journal 2020; K Watanabe, S Takesue, Journal of General Virology 1972; Fred Shafia, et al, Journal of Bacteriology 1964). They should consider testing this idea experimentally or properly citing those papers to summarize other important factors they might ignore.
Author Response
Author's Notes to Reviewer in the attachment.

Reviewer 2 Report
The authors isolate and describe the characteristics of a bacteriophage infecting Alicyclobacillus acidoterrestris, the first of its kind. As their interest lies in applying this virus to phage therapy, it is unfortunate that it is a prophage. However, the work is solidly performed and I have very few comments.
Line 278: I can not find any indication that phage size is inversely proportional to plaque size in that reference
Figure 2: As I understand, these growth curves where done in 10 replicates. Why no error bars then?
Line 367: Unless the bacteria were exposed to these temperatures, loss of pfu/ml due to temperature can not be indicative of increased lysogeny. Virus particles can not be stressed into switching to lysogeny.
Figure 6: Why is the phage itself not assigned to a host group (Firmicutes) and virus family (Siphoviridae)? Noting of course that Siphoviridae as a group have been dissolved. The authors could attempt a better classification using vContact2, but that is not strictly necessary for the publication of this paper.
Author Response

(The authors gave the same response as above.)

Round 2
Reviewer 1 Report
The authors answered all my questions. I have no further comments.